# Data from the Efficacy Study of From Here to There! A Dynamic Technology for Improving Algebraic Understanding

Journal of
open psychology data

DATA PAPER

ERIN OTTMAR

JI-EUN LEE

KIRK VANACORE

SIDDHARTHA PRADHAN

LAUREN DECKER-WOODROW

CRAIG A. MASON

*Author affiliations can be found in the back matter of this article

]u[ ubiquity press

## ABSTRACT

This paper provides information on datasets for the research project that examined the efficacy of three educational technologies including "From Here to There!", a research-based game for improving algebraic understanding. The dataset contains 4,092 7th-grade students' data collected through a randomized control trial conducted in 2020–2021 in a large school district in the U.S. The data comprises over 400 measures, including student demographics, assessments, and students' actions. All data is anonymized and stored on Open Science Framework (OSF) and available through a data-sharing agreement. Our data might be reused by researchers interested in students' algebraic learning in online learning environments.

CORRESPONDING AUTHOR:

**Erin Ottmar**

Worcester Polytechnic Institute, US

erottmar@wpi.edu

KEYWORDS:
Educational technology; Mathematics learning; Middle school; Randomized Control Trial; Problem-solving

TO CITE THIS ARTICLE:
Ottmar, E., Lee, J.-E., Vanacore, K., Pradhan, S., Decker-Woodrow, L., & Mason, C. A. (2023). Data from the Efficacy Study of From Here to There! A Dynamic Technology for Improving Algebraic Understanding. *Journal of Open Psychology Data,* 11: 5, pp. 1–15. DOI: https://doi.org/10.5334/jopd.87

# (1) BACKGROUND: AIMS OF THE PROJECT AND CONTEXT

## 1.1 AIMS OF THE PROJECT

The research project described in this paper tested the efficacy of an online interactive game, From Here to There! (hereafter, FH2T) compared to active control and two other educational technologies, ASSISTments and DragonBox 12+ (hereafter, DragonBox), across four intervention conditions on middle school students' algebraic understanding. Although these three technologies aim to improve students' mathematical understanding, they have different characteristics regarding instructional design, approaches, learning goals, and data types logged. We hypothesized that FH2T, an interactive game developed based on theories of perceptual learning and embodied cognition, may improve students' algebraic understanding above and beyond active control and the other two technologies.

Specifically, the overall goals of the project were:

1. To examine whether FH2T improves student achievement and growth more than DragonBox, ASSISTments-Immediate Feedback, and Active Control conditions.
2. To examine whether FH2T is more effective for some students than others, depending on prior achievement and student characteristics (i.e., gender, race/ethnicity, math anxiety, English as a Second Language (ESL), gifted, Early Intervention Program (EIP), etc.)
3. To explore plausible mechanisms by which FH2T leads to learning gains.
4. To advance basic and applied research in how perceptual learning, attention, and gesture-based interfaces help facilitate math reasoning in educational contexts.

The data reported here was collected through a Randomized Controlled Trial (RCT) conducted in 2020–2021 in a large school district in the U.S. The dataset contains information about over 4,000 7th-grade students, which comprises over 400 measures, including student demographics, assessments, and students' actions within three technologies across four intervention conditions. All data is anonymized and stored on Open Science Framework (OSF) and available through a data-sharing agreement.

## 1.2 CONTEXT

Algebra is considered a gatekeeper to students' further mathematics learning and academic success (Lynch & Star, 2014). Despite its importance, many middle school students often struggle with learning algebra. National Assessment of Educational Progress (Hussar et al., 2020) reported that only 34% of eighth graders were proficient in mathematics, and their performance has stagnated over the past years. Developing a deep conceptual and procedural understanding of algebra requires a greater focus on algebraic structures, abstract symbols, and mathematical relations such as equivalence. However, studies have found that middle school students have difficulties in (1) understanding the mathematical equivalence (Alibali et al., 2007; Booth & Davenport, 2013; Knuth et al., 2006) and (2) working with abstract symbols (or variables) representing unknown values (Bush & Karp, 2013; Star et al., 2015), and (3) perceiving underlying structure in algebraic equations (Ottmar & Landy, 2017).

In order to address these issues, Weitnauer, Landy, and Ottmar (2016) have designed and developed a research-based online interactive game, From Here to There! (FH2T; freely available on https://graspablemath.com/projects/fh2t) to improve students' algebraic understanding. FH2T uses cognitive and learning science theories (e.g., perceptual training, embodied cognition) and game design elements (e.g., rewards, challenge) to address many factors that lead to low proficiency. The objective of the game is to transform an algebraic expression into a mathematically equivalent but perceptually different goal-state expression (Figure 1).

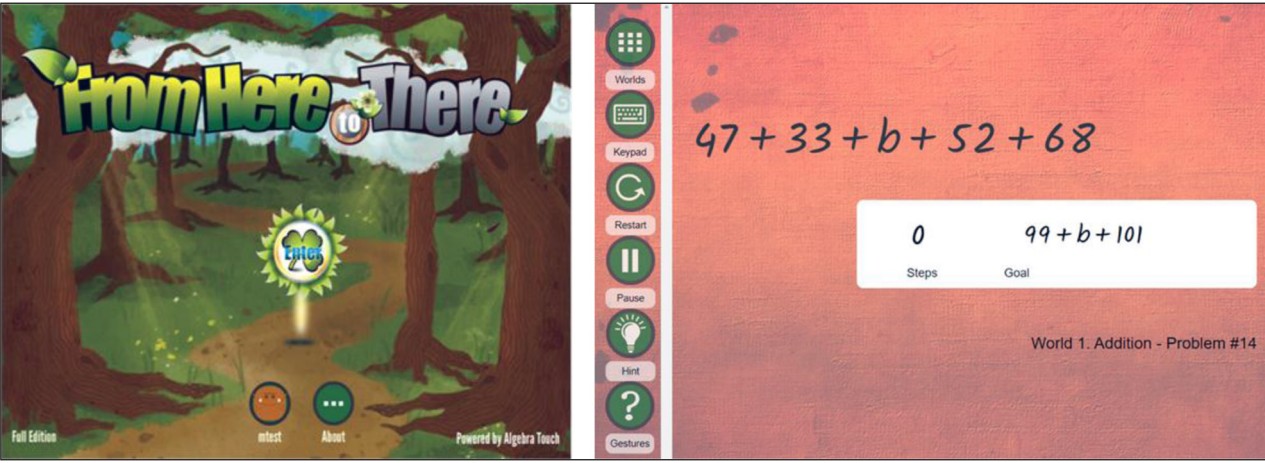

**Figure 1** FH2T start screen (left) and a sample problem in FH2T (right).

One of the key characteristics of FH2T is that the symbols and numbers in the game are made into movable physical objects, which enables students to dynamically manipulate and transform them on the screen using a series of gesture actions. In this way, students can easily identify underlying algebraic structures, think more flexibly, and realize that mathematical transformations are more dynamic than a static re-copying of lines. Our previous studies have found that FH2T is effective in improving elementary and middle school students' algebraic understanding (Hulse et al., 2019; Ottmar et al., 2012; 2015; Ottmar & Landy, 2017). However, the efficacy of the game had not been rigorously tested compared to other educational technologies under an RCT.

ASSISTments (https://new.assistments.org/) is a free online tutoring system that covers mathematical content well-aligned with traditional instruction and provides hints and immediate or delayed feedback on students' problem solving (Heffernan & Heffernan, 2014). It has different bookend aspects of FH2T; the problems in ASSISTments resemble those in mathematics textbooks, which are presented in a static format, and do not include perceptual features or gamified elements.

DragonBox (https://dragonbox.com/products/algebra-12; available on iOS and Android; provides free access for teachers) is a commercial, award-winning game-based application for practicing algebra. It was specifically selected as one of the comparisons over other game-based educational technologies because it has both similar and different characteristics to FH2T. Like FH2T, it is grounded in perceptual learning theories and uses gestures and multiple representations to introduce algebraic concepts to students. However, unlike FH2T, numbers or mathematical notations are hidden using pictures of dragons at the beginning of the game, and pictures are gradually replaced by algebraic symbols as the game proceeds. One of the main design principles of the game is that students should not perceive that they are doing math while playing the game so that it never teaches that the pictures represent math properties.

These similarities and differences across the three technologies (i.e., FH2T, ASSISTments, DragonBox) allowed our project to address some specific research questions of general interest productively. More detailed information on the background, research questions, and results of RCT can be found in our previous work (Decker-Woodrow, 2023).

## (2) METHODS

### 2.1 STUDY DESIGN

We conducted a student-level RCT comparing student learning from FH2T to two other technologies across three conditions: DragonBox, ASSISTments-Immediate Feedback (hereafter, Immediate Feedback), and ASSISTments-Active Control (hereafter, Active Control) conditions (Figure 2).

The technologies used, the math problems included, and the study procedures for the Immediate Feedback and the Active control conditions were identical. The only difference between the two conditions was the timing of the feedback. While the students in the Immediate Feedback condition were allowed to ask for hints and received correctness feedback during their problem-solving, the hints and feedback were only available after solving each problem for the students in the Active Control condition.

The students were randomly assigned to one of the four conditions, FH2T (40% of participants), DragonBox (20%), Immediate Feedback (20%), and Active Control (20%). Note that we assigned a higher percentage of students to the FH2T condition because the goal of this project was to test the efficacy of FH2T on students' algebraic learning in comparison to other educational technologies. Furthermore, note that the randomization occurred at the student level; therefore, students assigned to different conditions within the same class.

The study consisted of 13 sessions; four sessions for assessments (i.e., pre-, mid-, post-, and end-of-year assessments) and nine sessions for intervention (Figure 3).

All students took a 45-minute pre-assessment on their algebraic knowledge, math anxiety, and math self-

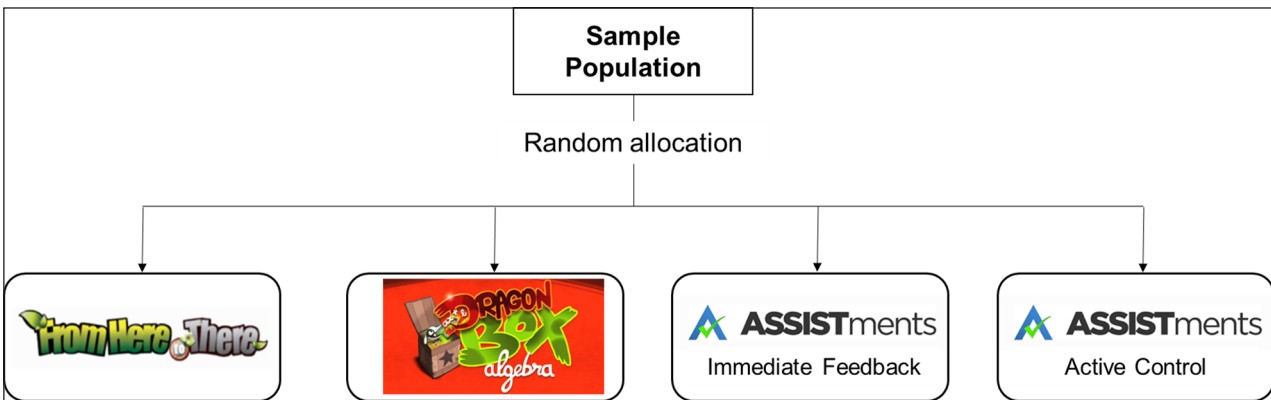

**Figure 2** Study Design of the FH2T Efficacy Study.

Ottmar et al. *Journal of Open Psychology Data* DOI: 10.5334/jopd.87

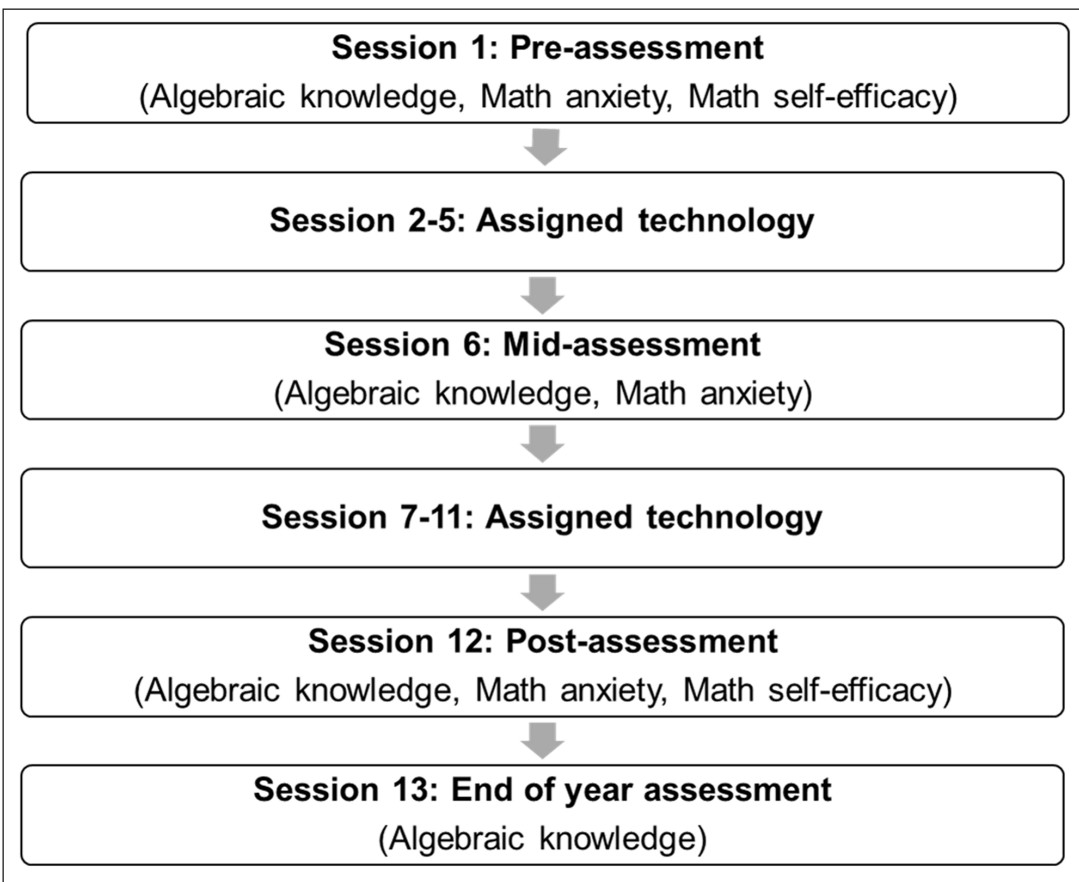

**Figure 3** FH2T Efficacy Study Procedure.

efficacy prior to intervention sessions on ASSISTments system. It is important to note that ASSISTments is designed to be used as not only an online tutoring system for teachers and students but also as a research platform to help researchers conduct RCT and collect student data. We also used ASSISTments in two different ways in this project: (a) as a platform for implementing the RCT (e.g., administering pre, mid, post, and delayed posttests; hereafter called "ASSISTments system" for distinction), and (b) as a technology used for the intervention (i.e., Immediate Feedback, Active Control). Then, students worked on their learning activities using their assigned technology (i.e., FH2T, DragonBox, Immediate Feedback, Active Control) for nine 30-minute sessions during their regular math classes. The students used their own devices, but the students who were in DragonBox condition and worked through the problems at school used tablets provided by our project team. In addition to the learning activities, students completed three additional common assessments (mid-, post-, and end-of-year assessments) that measured their algebraic knowledge or math anxiety during and after the intervention.

Given restrictions of physical distancing due to the COVID-19 pandemic, the school district offered students and their families a choice of classroom format (100% in-person or 100% asynchronous virtual academy) for the 2020-2021 school year prior to the start of the fall semester. Random assignment to study conditions occurred across formats, so the classroom format was not aligned to any one study condition. Regardless of students' classroom format, all study sessions (i.e., assessments and interventions) were administered online, and the students worked individually at their own pace using a device.

## 2.2 TIME OF DATA COLLECTION
The data was collected throughout an academic year, from September 2020 to April 2021. Note that this was during the COVID-19 pandemic when there were numerous learning disruptions in schools. To the best of course ability, we recorded student movement (i.e., transition from in-person to virtual or vice versa) to account for learning modality choices and classroom changes as a result of the pandemic.

## 2.3 LOCATION OF DATA COLLECTION
We collected data from 11 middle schools (10 in-person and one virtual academy) in one suburban school district in the southeastern United States.

## 2.4 SAMPLING, SAMPLE, AND DATA COLLECTION
The initial dataset included participant data from 4,343 7th-grade students from 190 mathematics classes in 11 middle schools (10 in-person schools and one virtual

academy). However, we excluded 251 students who were not enrolled in the school district at the point of the random assignment, which resulted in an analytical sample of 4,092 students. The number of teachers who taught these 4,092 students was 52. Table 1 represents student demographic information for the full sample and by intervention condition.

In order to test the primary research questions about the efficacy of FH2T in comparison to two other educational technologies across four conditions, we used a subset of the 4,092 students due to attrition. First,

we excluded the students ($n = 381$) in one school that opted out of the study before the pre-assessment due to concerns about COVID-19. We then eliminated 120 additional students in resource settings because these students were assigned to only FH2T and DragonBox conditions. Of the 3,591 students, we eliminated an additional 1,741 students who did not complete either the pretest ($n = 741$) or posttest ($n = 1,000$) assessments. Thus, the final analytical sample to test the efficacy of the interventions included 1,850 students (i.e., 4,092– 381–120–741– 1,000 = 1,850). Specific details on

| | FULL SAMPLE $N = 4,092$ (100%) | FH2T $N = 1,649$ (40.3%) | DRAGONBOX $N = 854$ (20.9%) | ASSISTMENTS-IMMEDIATE FEEDBACK $N = 795$ (19.4%) | ASSISTMENTS-ACTIVE CONTROL $N = 794$ (19.4%) |
|---|---|---|---|---|---|
| Gender | | | | | |
| Male | 2,133 (52.1%) | 867 (52.6%) | 451 (52.8%) | 400 (50.3%) | 415 (52.3%) |
| Female | 1,954 (47.8%) | 782 (47.4%) | 401 (47.0%) | 394 (49.6%) | 377 (47.5%) |
| Not reported | 5 (0.1%) | – | 2 (0.2%) | 1 (0.1%) | 2 (0.2%) |
| Race/Ethnicity | | | | | |
| White | 2,088 (51.0%) | 821 (49.8%) | 431 (50.5%) | 429 (54.0%) | 407 (51.3%) |
| Asian | 1,024 (25.0%) | 410 (24.8%) | 209 (24.5%) | 197 (24.8) | 208 (26.2%) |
| Hispanic | 617 (15.1%) | 270 (16.4%) | 140 (16.4%) | 98 (12.3%) | 109 (13.7%) |
| Black | 200 (4.9%) | 83 (5.0%) | 38 (4.4%) | 44 (5.5%) | 35 (4.4%) |
| Native American | 26 (0.6%) | 11 (0.7%) | 5 (0.6%) | 3 (0.4%) | 7 (0.9%) |
| Pacific Islander | 2 (0.1%) | – | 1 (0.1%) | – | 1 (0.1%) |
| Multi-racial | 135 (3.3%) | 54 (3.3%) | 30 (3.5%) | 24 (3.0%) | 27 (3.4%) |
| Accelerated math class | 720 (17.6%) | 283 (17.2%) | 147 (17.2%) | 146 (18.4%) | 144 (18.1%) |
| Gifted | 627 (15.3%) | 262 (15.9%) | 119 (13.9%) | 122 (15.3%) | 124 (15.6%) |
| EIP | 291 (7.1%) | 114 (6.9%) | 58 (6.8%) | 59 (7.4%) | 60 (7.6%) |
| IEP | 463 (11.3%) | 188 (11.4%) | 132 (15.5%) | 76 (9.6%) | 67 (8.4%) |
| ESL | 372 (9.1%) | 164 (9.9%) | 77 (9.0%) | 57 (7.2%) | 74 (9.3%) |
| Virtual | 1,618 (39.5%) | 650 (39.4%) | 331 (38.8%) | 319 (40.1%) | 318 (40.1%) |

**Table 1** Student Demographic Information by Condition ($N = 4,092$).

*Note*: EIP = Early Intervention Program is designed for students who need extra support to meet academic grade level, IEP = Individualized Education Program is designed for students with disabilities or special health care needs, ESL = English as Second Language.

processes of data cleaning can be found in our previous work (Decker-Woodrow, 2023) and our project webpage on Open Science Foundation (OSF; https://osf.io/r3nf2). Note that the data published on OSF included the entire sample of students regardless of their inclusion in the efficacy study analysis.

## 2.5 MATERIALS/SURVEY INSTRUMENTS

### 2.5.1 Materials
*FH2T*
FH2T consists of 14 worlds, and each world covers different mathematical topics, such as addition, multiplication, fraction, and division, with increasing difficulty. Each world contains 18 problems, and students must complete at least 14 problems to progress to the next world. Figure 4 represents the sample problem in FH2T.

The goal of this problem is to transform the start state (e.g., 16*29; Figure 4a) into the mathematically equivalent but perceptually different goal state expression in a white box (e.g., (30–1)*4*4; Figure 4a). In FH2T, the symbols and numbers are made into physical objects so that students can dynamically manipulate and transform them on the screen using a series of gesture actions (e.g., splitting, moving; Figure 4b-4e). Each gesture-action leading to a valid transformation is considered a step. Students are encouraged to transform expressions from a starting state to a goal state using more efficient strategies involving fewer steps. Three clovers are given (Figure 4f) if the student reaches the goal using the minimum required number of steps to solve the problem. See the project OSF page (https://osf.io/jne84) for the full list of 252 problems in FH2T, including the start state and goal state of each problem.

*ASSISTments*
The ASSISTments intervention (both Immediate Feedback and Active control conditions) for the current study was composed of nine mathematical sessions, each consisting of 24 to 39 multiple-choice or closed questions. The problems were adapted from three open-source middle-school mathematics curricula: Utah Math Project (2016), Illustrative Mathematics (2017), and Engage NY (2014).

The mathematical content covered in the problem sets in ASSISTments was well-aligned with Common Core State Standards (CCSS) in the U.S. as well as the topics covered in FH2T. Note that CCSS refers to a set of educational standards (i.e., academic expectations) for mathematics and English/language arts, and has been accepted by 46 states in the U.S. as of 2023. Figure 5 represents a sample problem in ASSISTments, and the full list of the problems in ASSISTments is provided on the project OSF page (https://osf.io/r3nf2, See the folder labeled ASSISTments problem sets).

*DragonBox*
Like FH2T, DragonBox also covers various algebraic concepts, such as addition, division, parentheses, and collection of like terms. DragonBox is comprised of 10 chapters, and each chapter contains 20 problems. Figure 6 presents sample problems in DragonBox. The goal of the game is to isolate the box containing a dragon to one side, which is equivalent to solving an equation for x. As shown in Figure 6a, numbers or mathematical notations are hidden at the beginning of the game using picture-based symbols, but students are gradually exposed to classical algebraic symbols as the game progresses (Figure 6c–6e). The full list of problems is not available because of its commercial license.

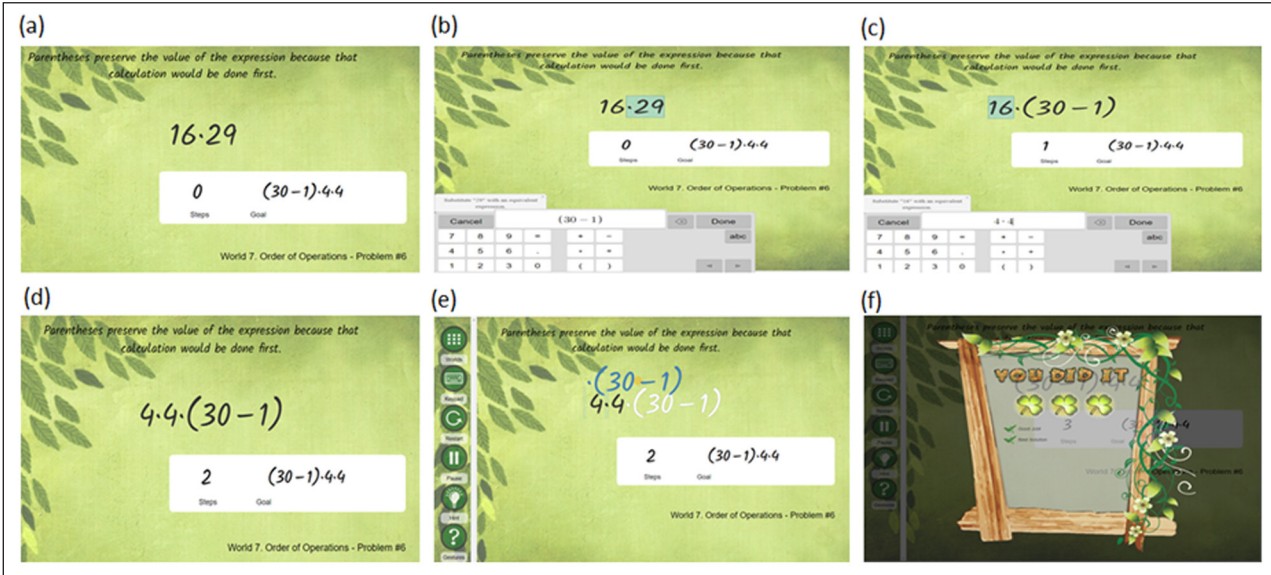

**Figure 4** A sample problem and students' actions in From Here to There!.

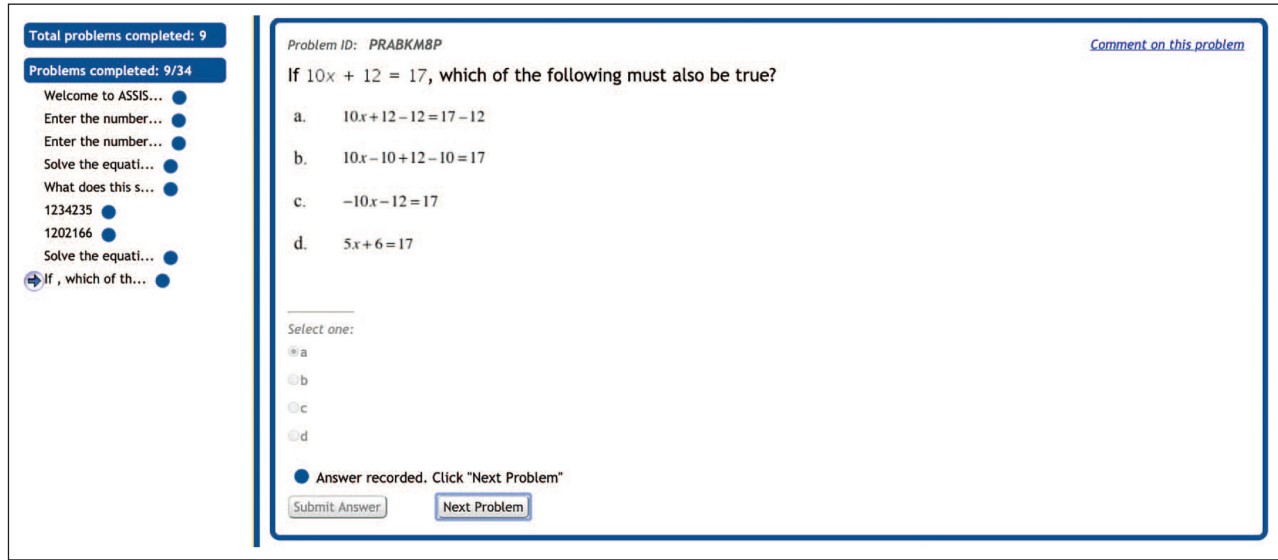

**Figure 5** A sample problem in ASSISTments.

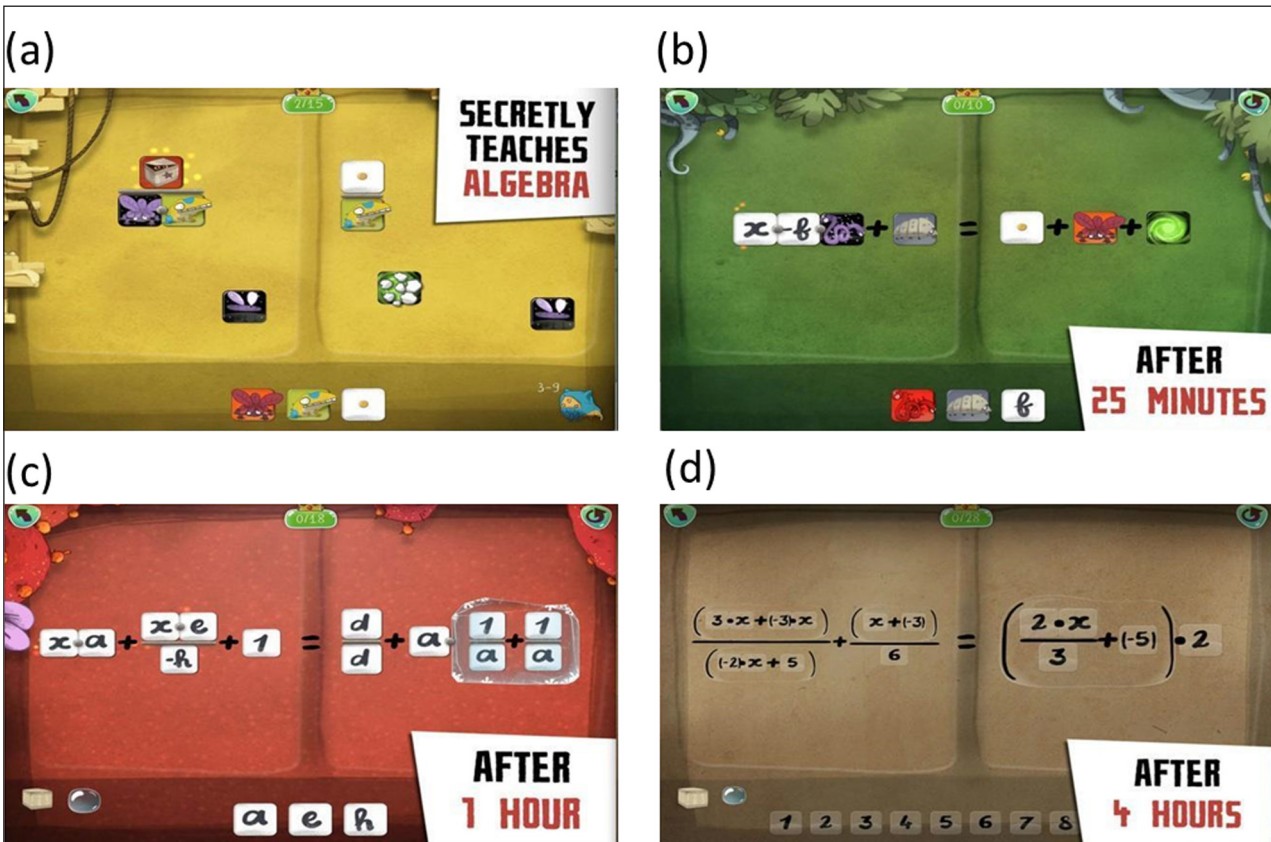

**Figure 6** Sample problems in DragonBox (used with permission by Kahoot!).

### 2.5.2 Survey instruments

*Algebraic Knowledge Assessment (pre, mid, post, and end-of-year assessments)*

We measured students' algebraic knowledge using ten items adapted from a previously validated measure (Star et al., 2015). We used isomorphic items that had the same problem structure but different numbers for assessments at four-time points. The assessment consisted of three sub-constructs, conceptual knowledge (math equivalence; 4 items), procedural knowledge (solving algebraic equations; 3 items), and flexibility (evaluating different strategies; 3 items). See the project OSF page for the full list of assessment items (https://osf.io/uenvg). All items were scored as correct (1) or incorrect/no attempt (0), and then we computed the sum of the item scores as pre-, mid-, post-, and end-of-year assessment scores, with ten being a perfect score. The inter-item reliability coefficients of the ten items were KR-20 = 0.74 at the pretest and KR-20 = 0.79 at

the posttest (Kuder-Richardson Formula; a reliability measure for binary variables).

### Perceptual Processing Skills

In addition to students' algebraic knowledge, we measured students' perceptual processing skills, which refers to students' ability to detect mathematically equivalent and nonequivalent expressions as quickly as possible (Bye et al., 2022). This task assessed students' perceptual processing with a short comparison of success on simple algebraic problems in which the physical spacing of the symbols has been manipulated to either match or mismatch the order of operations (Kirshner & Awtry, 2004; Landy & Goldstone, 2007). The task consisted of two parts with eight items each (see Figure 7 for sample items and the project OSF page for the full list of items: https://osf.io/r47ev). For each problem in Part 1, students determined whether two expressions were equivalent or inequivalent. In Part 2, students saw a target expression with six options and selected the option that was equivalent or not equivalent to the target. We recorded the accuracy and response time on each item.

### Math anxiety and Math self-efficacy

To measure math anxiety, we used nine items adapted from the Math Anxiety Scale for Young Children-Revised (Ganley & McGraw, 2016; Cronbach's α = .87). The scale comprises three sub-construct that measured students' negative reactions (3 items), numerical inconfidence (3

items), and worrying (3 items). Students rated how well each item described their feeling towards mathematics on a four-point scale (No = 0, Not really = 1, Kind of = 2, Yes = 3). To assess math self-efficacy, we used five items adapted from the Academic Efficacy subscale of the Patterns of Adaptive Learning Scale (Midgley et al., 2000; Cronbach's α = .82). Students rated how often they felt a certain way about math on a six-point scale (Never = 0; Very rarely = 1; Rarely = 2, Often = 3, Very often = 4, Always = 5). Scores were then averaged to create the math anxiety and self-efficacy composite. See the project OSF page (https://osf.io/rq9d8) for the full list of math anxiety and self-efficacy items. Table 2 lists the survey instruments and the locations and file names of each instrument on the project OSF page.

### 2.6 QUALITY CONTROL

Our dataset meets data quality standards with regard to several aspects, such as quantity, quality, utility, and accuracy. First, our data contains a large amount of student data (*N* = 4,092) collected through an RCT conducted in a large school district in the Southern U.S. over nine intervention sessions across a school year. For successful data collection and completion of the RCT, we conducted a pilot study in the same school district with a smaller sample (*N* = 475) during the 2019–2020 school year. The results of the pilot study are reported in our previous work (Chan et al., 2022a). In addition, several manuscripts were published from this pilot data (Iannachionne et al., 2022; Lee et al., 2022a; 2022b;

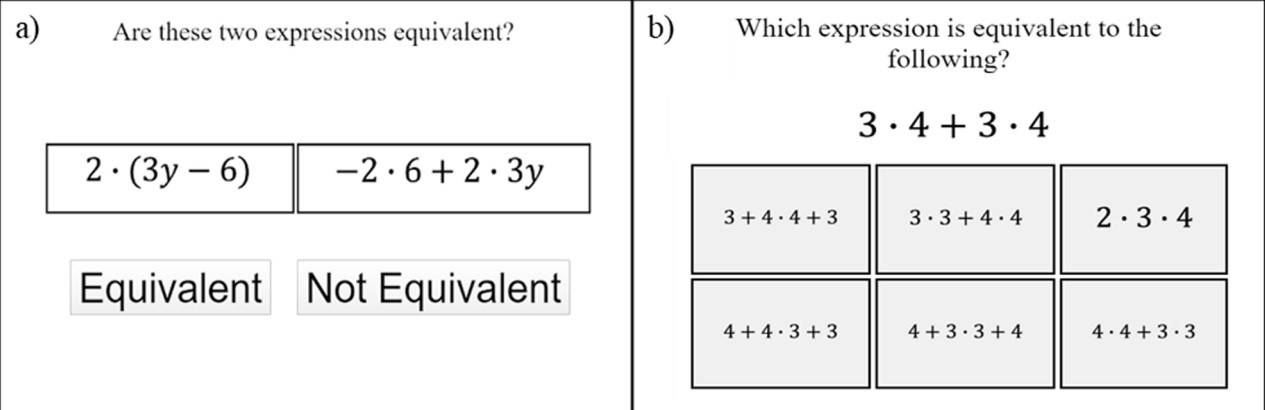

**Figure 7** Sample items in the first **(a)** and second **(b)** part of the Perceptual Math Equivalence Task.

| MEASURES | NUMBER OF ITEMS | LINK TO THE FILES | LOCATION AND THE FILE NAME ON THE OSF PAGE |
|---|---|---|---|
| Algebraic Knowledge | 10 | https://osf.io/uenvg | 1. Problem list and assessment items/Assessment items.pdf |
| Perceptual Processing Skills | 16 | https://osf.io/r47ev | 1. Problem list and assessment items/Perceptual sensitivity task_items.pdf |
| Math anxiety and Math self-efficacy | 14 | https://osf.io/rq9d8 | 1. Problem list and assessment items/Math_Anxiety_Math Self-Efficacy_items.pdf |

**Table 2** The list of the survey instruments and the location of the instrument files.

2022c), demonstrating the usefulness of this data for educational and cognitive psychology research.

The complete data is composed of multiple datasets collected from four intervention conditions, including various types of student data (i.e., logs, text, demographics, proximal math assessments at four-time points, and state-standardized math assessments at two-time points) with several different hierarchies (i.e., action and problem-level logs, assignment-level, overall-level aggregations). In terms of quality and accuracy, the data was pre-processed and de-identified by our research team. We conducted error detection and correction to ensure the accuracy of the datasets. We also created metadata and detailed data dictionaries, including descriptions of each dataset and variable type for easy access and use by other researchers and practitioners.

## 2.7 DATA ANONYMIZATION AND ETHICAL ISSUES

This study obtained Institutional Review Board (IRB) approval, and the primary institution oversees the study under the "research conducted in established educational settings" exemption. Parents were provided with a letter about the study and could opt out of their child from participation. There is also a shared Memorandum of Understanding (MOU) agreement (i.e., mutual agreement between two parties) from the school district and all research partners, and our IRB has approved a data-sharing process to allow access to this rich de-identified data for others outside of the initial research team. The PIs have fully de-identified these files and made them available as separate, fully de-identified files on OSF to the public for research purposes with a signed DSA. While the data is not able to be fully open and has unrestricted access through a CCO license, instructions for how the public can access the files and documentation through the DSA process are publicly available on OSF.

## 2.8 EXISTING USE OF DATA

Using this data, over ten research articles have been published, over 20 presentations have been made at national and international conferences, and approximately ten more research articles are under review as of December 2022. Below is a list of selected publications, and a full list of published papers and presentations from this data is available online at https://tinyurl.com/szvnwndn.

### Selected Research Articles

- Decker-Woodrow, L., Mason, C. A., Lee, J. E., Chan. J. Y. C., Sales, A., Liu, A., Tu, S. (2023). The impacts of three educational technologies on algebraic understanding in the context of COVID-19. *AERAOpen*.
- Chan, J. Y. C, Lee, J., Mason, C., Sawrey, K., & Ottmar, E. (2022a). Equivalence task in From Here to There!: A digital algebraic notation system impacts conceptual understanding in middle school mathematics. *Journal of Educational Psychology.* https://doi.org/10.1037/edu0000596
- Chan J. Y., Ottmar, E., & Lee, J. E. (2022b). Slow down to speed up: Longer pause time before solving problems relates to higher strategy efficiency. *Learning and Individual Differences, 93,* 102109. https://doi.org/10.1016/j.lindif.2021.102109
- Lee, J. E., Chan, J. Y. C., Botelho, A., & Ottmar, E. (2022a). *Does slow and steady win the race?:* Clustering patterns of students' behaviors in an interactive online mathematics game. *Educational Technology Research and Development.* https://doi.org/10.1007/s11423-022-10138-4

# (3) DATASET DESCRIPTION AND ACCESS

This data provides information about the study at multiple levels from metadata on student demographics, to aggregated data on student performance, to item and action level data on student behaviors within each program. We have organized that data such that multiple levels of analysis are possible, including student, class, item, and action analyses.

## 3.1 ACCESSING REPOSITORY

The data are located on the OSF website and can be accessed by filling out the data sharing agreement found on the From Here To There Efficacy Study repository (DOI: 10.17605/OSF.IO/R3NF2; https://osf.io/r3nf2). This is the location of the initial data-sharing agreement files, study description, and data dictionaries. In order to gain access to the actual files, researchers will need to complete the data-sharing process and send their signed agreements to the project team. Once that request process is completed, a link to the data files will be sent to the researcher.

## 3.2 DATA TABLES (OBJECT/FILE NAME)

The data are organized into five sections of tables totaling 17 tables. The tables within each section are described in Table 3. The *student tables* include data on student demographics, attendance, roster, and study program usage fidelity. Much of the data in this section was provided by the students' school district. The data on student fidelity were collected within the ASSISTments system for all intervention conditions as the ASSISTments system used a portal through which students accessed other programs that logged students' time use of their assigned programs. The *assessment tables* contain data on student performance on both the assessments administered as part of the study and the student performance on the state-administered exams.

The *FH2T* and *ASSISTments tables* contain log-file data for student progress and performance on their respective conditions. This log file data was aggregated at different levels (action, problem, student, etc.) to allow for different levels of analysis. The DargonBox tables contain session-level data that came from self-reports, in which students documented their performance and progress during each session/assignment, and the final progress and performance data that were manually extracted from the game devices. We did not have access to log file data for the DragonBox conditions, so the tables do not include granular problem or actionlevel data.

Each table contains unique anonymous identifiers for each student or problem. These identifiers are documented in the data dictionary described in Section 3.9. Using these identifiers, a researcher can merge multiple layers of data for specific analyses. For example, if a researcher is interested in evaluating whether students from different demographic groups used the programs differently, they could merge the student_demo and student_fidelity tables using the student identifier (i.e., StuID). Alternatively, if a researcher desires to understand the relation between students' performance on different types of problems in FH2T and their growth in algebraic knowledge across the study, they would need to combine three data tables. They would merge the "assess_student" table to the "fh2t_student_problem" table using the student identifier (i.e., StuID) to connect student performance on the assessment with their performance on individual problems. Then they would merge the "fh2t_student_problem" data to the problem "fh2t_problems_meta" using the problem identifier (i.e., problem_id) to connect information about the problems to the students' performance data. These mergers would create a data table containing the student's assessment data, the student performance on each of the problems they attempted, along with information about those problems.

### 3.3 DATA TYPE
The dataset provided a variety of data on students assessments, demographics, progress and performance within programs. There are four types of programmatic data: raw clickstream data, pre-processed data, self-report data, and data recorded manually from devices. For the FH2T and ASSISTments, both raw clickstream data and pre-processed data are available. The raw clickstream data allows for action-level analysis, including each students' raw responses, errors, hint usage, etc. The pre-processed data is aggregated from the action data at different levels – problem and student – allowing researchers to easily access data at the level that aligns with their research questions. For DragonBox, we do not have access to the action-level data. Instead, we have self-reports of student progress within each session, and final usage and progress are recorded

manually from students' devices. For assessment data, the dataset includes problem-level data response data and agitated data on all pre, mid, post, and distal (i.e., end-of-year) assessments as well as secondary data on students' standardized test assessments provided by the district. For fidelity data, we pre-processed data on students' assignment usage throughout the study. The dataset also includes secondary data provided by the districts on students' demographics, rosters, and attendance. Figure 8 depicts the data structures of three educational technologies (FH2T, ASSISTments, DragonBox) and examples of the metrics.

### 3.4 FORMAT NAMES AND VERSIONS
The data are available in two formats: a series of comma-separated value files (.csv) and tables in an SQLite relational database (.db) that can be downloaded and used on a personal computer. The two formats mirror one another, with each csv file having a corresponding and equivalent table in the database. If the researcher is using data from multiple tables/files, and aggregating data at different levels students/assignment/problem using querying the SQLite database through the free SQLite Studio application may be preferable to loading and manipulating multiple files in statistical software.

### 3.5 LANGUAGE
The data is stored in American English.

### 3.6 LICENCE
The data description and request procedures and the full data have been published on two OSF pages without a CC0 licence (i.e., no copyright reserved). As mentioned earlier, the full data cannot be unrestricted due to our institution's IRB and the MOU with the school district but is open to the public on OSF with a signed DSA. The DSA process is described on the project OSF page and is overseen by the project PI at the primary institution.

### 3.7 LIMITS TO SHARING
This dataset is available to researchers who have completed the Data-Sharing Agreement (DSA) process with the Principal Investigator (PI)'s University. The DSA is a requirement of our institution's IRB as well as the signed MOU with the school district. As student-level data is covered under Family Educational Rights and Privacy Act (FERPA) law in the U.S., our institution requires researchers who want to use the data to sign an agreement not to share the data with anyone else. Thus, researchers who wish to use the data must fill out the two data-sharing agreement forms provided on our OSF page (https://osf.io/r3nf2) and send them along with their Collaborative Institutional Training Initiative (CITI) training certificate (Human Subjects in Social & Behavioral Research course) to the PI of the project team. Note that CITI (https://about.citiprogram.org/en/homepage) is a

Ottmar et al. *Journal of Open Psychology Data* DOI: 10.5334/jopd.87

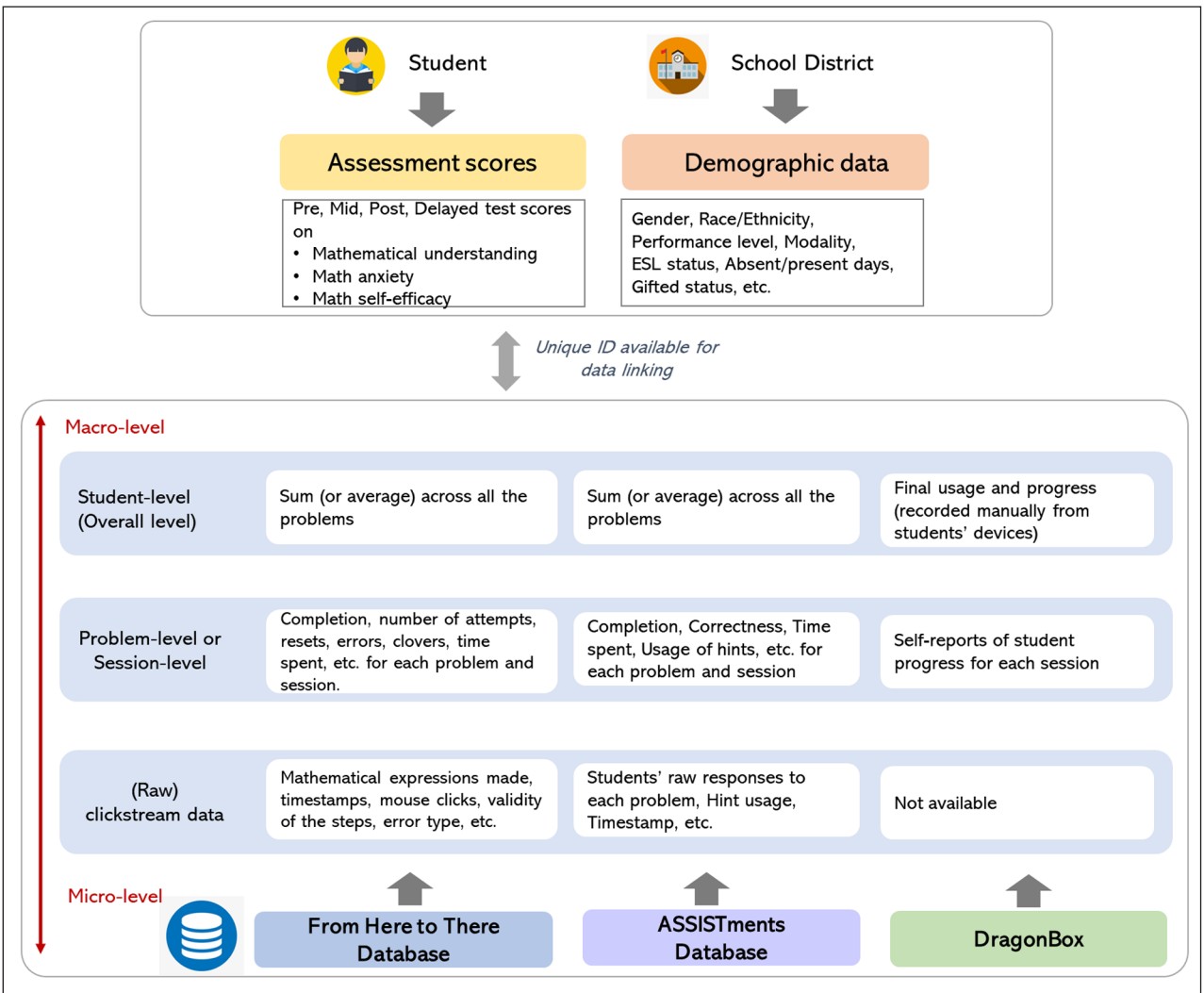

**Figure 8** Data structures of three educational technologies and examples of the metrics.

training program for the protection of human subjects in research. The CITI program is available to anyone aged 18 years and older, and the completion of the training will take approximately 2–3 hours. For researchers who reside in a country where the CITI program is not available, we also accept ethics training certifications in their country. Once researchers complete and return the DSA to the project team, they will receive access to the shared de-identified datasets and documentation. The estimated processing time is 2 to 3 business days.

### 3.8 PUBLICATION DATE

Public DSA Request Processes and Forms were added (09/07/2022).

DSA Approved dataset was added (09/26/2022).

### 3.9 FAIR DATA/CODEBOOK

The dataset meets FAIR data standards by being findable, accessible, interpretable, and reusable (Wilkinson et al., 2016).

***Findable:*** The metadata are easily findable on OSF, and the full dataset can be easily accessed by submitting

our data sharing agreement, which is on the OSF page. The metadata for our dataset are assigned a globally unique identifier DOI 10.17605/OSF.IO/R3NF2 (F1). The data include rich metadata, which are described in Table 3 and in more detail in the Data Dictionary available on OSF (F2). The metadata available on OSF is described in separate files (F3). The data are indexed in a searchable form through an SQL database (F4).

***Accessible:*** The data are accessible through a clickable link on OSF at https://osf.io/r3nf2 (A1), and the protocol for accessing data is free and universally open-sourced (A2) through the authentication and authorization of our DSA protocol (A3). Metadata will remain on OSF even after district-level data is no longer available due to the agreements in the MOU (A4).

***Interpretable:*** We provide documentation to ensure that the data are interpretable, including an extensive study description and a robust data dictionary. The data are provided in broadly used and accessible format (csv and database) through a commonly used program language (SQL) with a well-defined structure mapping metadata to data through the relational database and

| CSV | CONTENTS | STRUCTURE |
|---|---|---|
| **student** | *Note: Fully de-identified. | |
| **student_demo** *(student_demo.csv)* | Data provided by the district on students' demographics (race/ethnicity, gender, IEP, etc.) | one row per student |
| **student_roster** *(student_roster.csv)* | Data provided by the district on students' school, teacher, class, impersonal or virtual status and movement between status during the year of the study. Data on students' randomization and treatment status. | one row per student |
| **student_attendance** *(student_attendance.csv)* | Data provided by the district on students' attendance for fifth, sixth, and seventh grade. | one row per student |
| **student_fidelity** *(student_fidelity.csv)* | Data on which of the 11 study assignments students started and completed. These data were captured by the ASSISTments system. | one row per student |
| **assessment** | | |
| **assess_student** *(assess_student.csv)* | Student performance data on pre, mid, post, and end-of-year assessments were captured in the ASSISTments system. Student state test scores provided by the district for fifth and seventh grade. (Note that the state test was not administered in 6th grade due to the COVID-19 pandemic.) | one row per student |
| **assess_student_problem** *(assess_student_problem.csv)* | Item level data on pre, mid, post, and end-of-year assessments captured in the ASSISTments system. | one row per student (wide) |
| **From Here To There (FH2T)** | | |
| **fh2t_problems_meta** *(fh2t_problems_meta.csv)* | Metadata on FH2T problems (problem equations and solutions, number of optimal steps, problem sequence, hints and instructional text, etc.) | one row per problem |
| **fh2t_student_action_logs** *(fh2t_student_action_logs.csv)* | Student action log data student (actions taken, response time, whether the action was valid mathematically or a mistake, the state of the equation before and after the action, etc.) | one row per student action (long) |
| **fh2t_student_problem_attempt** (fh2t_student_problem_attempt.csv) | Aggregated data on student problem attempts within FH2T (number of steps taken, number of hints requested, number of errors, etc.) | one row per student per problem attempt (long) |
| **fh2t_student_problem** (fh2t_student_problem.csv) | Aggregated data on student problems within FH2T including overall data on the problem (total time, number of attempts, number of replays, etc.) and data on students first, last and best attempt at the problem (number of steps taken, number of hints requested, number of errors, etc.) | one row per student per problem (long) |
| **fh2t_student** (fh2t_student.csv) | Aggregated data on student progress/performance within FH2T (Number of problems attempted, attempts per problem, total hints requested, etc.) | one row per student |
| **ASSISTments** | | |
| **assist_problems_meta** (assist_problems_meta.csv) | Metadata on ASSISTments problems (problem order, problem type, curriculum for which the problems were derived, etc.) | one row per problem part |
| **assist_student_action_logs** *(assist_student_action_logs.csv)* | Student action log data within ASSISTments (actions, response times, responses, etc.) | one row per student action (long) |
| **assist_student_problem** *(assist_student_problem.csv)* | Aggregated data on student problem attempts within ASSISTments (number of attempts, number of hints, accuracy, response times, etc.) | one row per student per problem (long) |
| **assist_student** *(assist_student.csv)* | Aggregated data on student progress and performance in ASSISTments (number of problems, total number of hints, accuracy, etc.) | one row per student |
| **DragonBox** | | |
| **dragon_student** (dragon_student.csv) | Aggregated data on student progress DragonBox (from devices) | one row per student |
| **dragon_session** *(dragon_session.csv)* | Self-report data on usage and progress in each session | one row per student (wide) |

**Table 3** Descriptions of each table.

data dictionary (I1). Documentation is easily findable through OSF, including unique identifiers (I2), and explanations of the associations between variables (I3).

**Reusable:** To ensure that the data are reusable, we provide rich metadata in relational database format

that has been reviewed by a team of research scientists, graduate researchers, and undergraduate research assistants. The data are presented in well-established file formats (CSV and SQL) and documentation in PDF and data table formats (R3). The data can be used only once a Data

Sharing Agreement has been signed and submitted to the project PI. Only those who complete this DSA process with the host institution can have access to the data (R1). The OSF page includes data on how to cite the data (R2).

# (4) REUSE POTENTIAL

This project may have significant implications for student math achievement across further grade levels, which rely on algebraic notation as a foundational representation. In addition, the knowledge gained through this project will benefit other researchers, mainly in cognitive science and math education fields. These data might be reused by researchers, students, and educational practitioners interested in middle school students' algebraic problem-solving processes and strategy use, mathematics learning, behavioral engagement in online learning environments, as well as educational big data analytics. This rich dataset was collected completely online across a school year within three well-established educational technologies and learning platforms during the pandemic. We expect that this dataset will provide researchers, students, and educational practitioners with rich learning and research opportunities to explore various types of educational big data (e.g., logs, assessments, demographics) collected from different types of educational technologies.

Using the dataset, our team has conducted research on (1) examining the efficacy of technologies from the RCT (Chan et al., 2022a; Decker-Woodrow, 2023), (2) comparing students' algebraic problem-solving or behaviors between two educational technologies (Chan et al., in prep; Iannachionne et al., 2022), (3) exploring students' algebraic problem-solving processes (e.g., pause time before solving problems) (Chan et al., 2022b) or behavioral patterns in FH2T (Lee et al., 2022a; 2022c), and (4) investigating the effects of algebraic problem structure (e.g., numbers, proximal grouping of numbers) on student learning (Lee et al., 2022b).

Although extensive research has been carried out using the dataset by our research team, our previous studies have not dealt in much detail with (1) students' mathematical misconceptions or errors, (2) differences between in-person and virtual students in their learning, and (3) associations between algebraic knowledge and state assessment scores. In terms of analytics methods, further research applying more sophisticated and advanced analytics methods, such as sequential pattern mining or longitudinal analyses, would be worthwhile. In sum, further research using data at different levels from our dataset would be of great help in better understanding middle school students' algebraic problem-solving processes, strategy use, mathematical misconceptions and learning, and behavioral engagement in online learning environments.

## ACKNOWLEDGEMENTS

Kathleen and Teachers
Avery Closser, Postdoctoral Research Scientist, Purdue University
Shifen Tu, Professor, University of Maine- Orono
David Brokaw, Graspable Inc.
David Landy, Netflix
Robert Goldstone, Professor, Indiana University-Bloomington
Anthony Botelho, Assistant Professor, University of Florida
Chris Donnelly, Worcester Polytechnic Institute
Daniel Manzo, Graduate Research Assistant, Worcester Polytechnic Institute
Katharine Sawrey, Tufts University
Vy Ngo, Graduate Research Assistant, Worcester Polytechnic Institute
Aravind Stalin, Graduate Research Assistant, Worcester Polytechnic Institute
Kathryn Drzewiecki, Graduate Student, Worcester Polytechnic Institute
Chloe Byrne, Undergraduate Student, Worcester Polytechnic Institute
Cindy Trac, Undergraduate Student, Worcester Polytechnic Institute
Yveder Joseph, Undergraduate Student, Worcester Polytechnic Institute
Hailey Anderson, Undergraduate Student, Worcester Polytechnic Institute
Justin Roberts, Undergraduate Student, Worcester Polytechnic Institute
Janette Jerusal, Undergraduate Student, Worcester Polytechnic Institute
Todeyon Yann Somasse, Undergraduate Student, Worcester Polytechnic Institute
DragonBox, Kahoot!

## FUNDING INFORMATION

The research reported here was supported by the Institute of Education Sciences, U.S. Department of Education, through an Efficacy and Replication Grant (R305A180401; 2017–2023) to Worcester Polytechnic Institute. The opinions expressed are those of the authors and do not represent views of the Institute or the U.S. Department of Education.

## COMPETING INTERESTS

Erin Ottmar was a designer and co-developer of From Here to There! The other authors declare no conflicts of interest associated with the publication of this manuscript.

## AUTHOR CONTRIBUTIONS

- Erin Ottmar: Funding acquisition, Project administration, Supervision, Writing original draft, Review and Editing.
- Ji-Eun Lee: Data curation, Writing original draft, Review and Editing.
- Kirk Vanacore: Data curation, Writing original draft, Review and Editing.
- Siddhartha Pradhan: Data curation.
- Lauren Woodrow-Decker: Validation, Review.
- Craig A. Mason: Methodology, Formal Analysis, Review.
- Furthermore, we would like to thank our colleagues involved in this project for their assistance in preparing the manuscript.
- Neil Heffernan, Co-PI, Worcester Polytechnic Institute
- Erik Weitnauer, Graspable Inc.
- Jenny Yun-Chen Chan, Assistant Professor, The Education University of Hong Kong
- Allison Liu, Postdoctoral Research Scientist, Worcester Polytechnic Institute
- Barbara Booker, Project Coordinator, Independent Consultant

## AUTHOR AFFILIATIONS

**Erin Ottmar** orcid.org/0000-0002-9487-7967
Worcester Polytechnic Institute, US

**Ji-Eun Lee** orcid.org/0000-0001-8521-8997
Worcester Polytechnic Institute, US

**Kirk Vanacore** orcid.org/0000-0003-0673-5721
Worcester Polytechnic Institute, US

**Siddhartha Pradhan** orcid.org/0009-0004-9977-1442
Worcester Polytechnic Institute, US

**Lauren Decker-Woodrow** orcid.org/0000-0001-9404-567X
Westat, US

**Craig A. Mason** orcid.org/0000-0002-9690-1692
University of Maine, US

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

## PEER REVIEW COMMENT

*Journal of Open Psychology Data* has blind peer review, which is unblinded upon article acceptance. The editorial history of this article can be downloaded here:

- **PR File 1.** Peer Review History. DOI: https://doi.org/10.5334/jopd.87.pr1

**TO CITE THIS ARTICLE:**
Ottmar, E., Lee, J.-E., Vanacore, K., Pradhan, S., Decker-Woodrow, L., & Mason, C. A. (2023). Data from the Efficacy Study of From Here to There! A Dynamic Technology for Improving Algebraic Understanding. *Journal of Open Psychology Data,* 11: 5, pp. 1–15. DOI: https://doi.org/10.5334/jopd.87

