## [Peer Review History. · Journal of Open Psychology Data]

Peer Review Reports for “Data from the Efficacy Study of From Here to There: A Dynamic Technology for Improving Algebraic Understanding”

Dear Erin Ottmar, Ji-Eun Lee, Kirk Vanacore, Siddhartha Pradhan, Lauren Decker-Woodrow, Craig A Mason,

After review, we have reached a decision regarding your submission to Journal of Open Psychology Data, "Data from the Efficacy Study of From Here to There: A Dynamic Technology for Improving Algebraic Understanding". Our decision is to request revisions of the manuscript prior to acceptance for publication.

The full review information is included at the bottom of this email. Please note that there may also be a copy of the manuscript file with reviewer comments available once you have accessed the submission account. We ask you to please consider the following issues and revise the file accordingly.

I am very grateful for two very rapid and in-depth reviewers from two expert reviewers. In summary, both reviewers had extremely positive comments about the conscientious nature of the production of your work, and the potential contribution of the work, so many congratulations. As we are the Journal of Open Psychology Data, we do have to be very careful about data access etc and both reviewers commented on their inability to see the data due to the range of processes required to access. I therefore ask that you make some amendments to your work to provide a more comprehensive account to the processes required to gain access to the data, potential barriers and routes to overcome these, and an explanation which is not US-centric. They also have a number of other useful comments which I encourage you to action. As such, I request you to resubmit this work, making it clear on the revised document where changes have been made (e.g. using tracked changes, coloured text, etc.). I look forward to receiving this revised document and I will endeavor to action it as promptly as possible!

Instructions for how to resubmit your article online are pasted below. Please ensure that your revised files adhere to our author guidelines, and that the files are fully proofed prior to upload. Please also include a revised version of your article with 'tracked changes', adding comments where appropriate, to indicate the revisions made, in addition to a document outlining how you have responded to the reviewers' requests.

If you have trouble processing the revisions, our Help Center (<https://help.u-community.io>) or downloadable PDF (<https://bit.ly/Author-Guide-OJS-3>) may be able to help. If not, please get in touch and we'll be happy to help.

Please also ensure that all copyright permissions have been attained for any figures/tables you have included.

Please could you have the revisions submitted with four weeks. If you cannot make this deadline, please let us know as early as possible but we always prioritise it being correct not quick!

Kind regards,

Dr Thomas Rhys Evans

Reviewer A:

Recommendation: Revisions Required

Disclosure: It is the first time I am reviewing a data paper, so even though I have a bit of experience with research data management and data sharing, I am not always fully confident in my judgments. Therefore, I am sharing my observations and conclusions to draw editor's attention to some points they may find potentially worth looking into.

This manuscript presents an extensive dataset from a randomised controlled trial testing efficacy of the FH2T game designed to improve arithmetic understanding in children. This intervention is compared to three different control groups.

I think this is an extensive and potentially very useful dataset, which could be of a great interest to the field. The paper also seems to fulfil the reviewing criteria of the JOPD so my evaluation is generally positive. However, I have some reservations, which I would like the editor to look at and the authors to consider.

Accessibility – I would leave it up to the editor, to decide whether data sharing upon data sharing agreement (and the need to justify the data request – see below) constitutes the acceptable level of access as for the journal requirements. I am not sure how to interpret the guidelines in the instructions for reviewers in this case (i.e., whether this is the “moderate access” or not). While reviewing the ms I could not access the data myself, as I did not engage with filling in the data sharing agreement (I do not think it should be reviewer's job). I went through the materials on the OSF page and found information that a copy of the CITI certificate is required from the researchers requesting access to the data. I am sorry for my ignorance here, but not being involved in US-American system, I have never heard of CITI Certificates, and to my knowledge they are not very widespread in Europe (based on experiences from Poland, Germany, and the UK). This is also not mentioned in the ms. CITI certificates should be also explained at least a bit, providing information whether non-US researchers can obtain them and whether this is associated with additional time / costs. The link in the data sharing request leads to general CITI website, which seems to have quite extensive training offer.

If it is quick and easy to obtain – please provide this information to reassure your readers that this will not be a barrier for them to access the data. It may also be worthwhile to mention in the ms how the data sharing process looks like in terms of how the request is reviewed and what is the approximate time needed for the request to be processed. I am also unsure whether it is necessary for the researchers to provide justification for data request. (Section: “Please briefly explain your intention for using the data.” in the data sharing request).

What is FERPA and MOU mentioned in the forms. Again – this might be obvious for US-based researchers, but I think it should not be taken for granted for researchers from outside the US.

Easiness to recreate the data – on top of detailed information on problem lists etc. it would be good if the authors provided information on conditions of use of the games / computer programs being used in the study. Are they commercial / freely available / what would be the conditions if other researchers are willing to use these instruments? Similar, for the Editor to look at – is not sharing the item level data from the DragonBox a problem? I suppose that it is not, but again – it may be that my interpretation is faulty.

Table 1. In the note the authors explain abbreviation of EIP and IEP. Would be good to add some further explanation / reference to non-us readers whether this is a specific term referring to the US educational system or just generic support provided to these children. What is the main difference between the two?

p.9 Term “Common Core Standards” also to my understanding refers to specific entity in US educational system and may not be obvious to non-US researchers.

p.13 description on how to combine different datasets is a bit dense and I think it would benefit from adding some more details and explanations.

Table 2 – seems to be mapping somehow on the folder “2. Data Dictionary” on the OSF. However, it would be good to also mention names of specific files in this folder, where the reader will find explanation what each column stands for in a greater detail. Sometimes it is a bit confusing. Would be also good to double check, whether everything is covered in files in this folder. For instance, file “2.3 Assessment.pdf” on the OSF at least to my reading only mentions summary data of the assessments, but not coding / variable names of single items of the assessment instruments, as described in the row “`assess_student_problem`” of Table 2.

“ASSISTments system” in Table 2 refers both to one of the interventions and to platform used to collect data in Assessment part of the RCT. Please be more specific here to avoid potential confusion. This is briefly mentioned in section 3.2 but may be useful to mention it also in the section on data gathering.

Table 2: section ASSISTments: last row potentially has a typo in the first column, stating “`assist_student`” and “`assist_student.csv`” -> I think there is one “s” too much.

Section 3.6

I do not fully understand what “OSF pages without a CC0 license” stand for? That no licence has been assigned (i.e., “all rights reserved”)?

As stated above, I did not engage with data sharing request (I do not think it should be required from the reviewer, and do not have the CITI certificate), so I cannot comment on the state of the actual datafiles and their suitability. In my opinion these should be supplied in some streamlined form for purpose of the review of the manuscript.

Signed: Krzysztof Cipora, Loughborough University

Reviewer C:

Recommendation: Accept Submission

Thank you for the opportunity to read and review the data paper: "Data from the Efficacy Study from Here to There: A Dynamic Technology for Improving Algebraic Understanding." This paper provides a thorough and detailed description of an individual student-level randomized control trial of three different educational technologies vs. an active control. This study examined the effects of these technologies on 4,000 students' algebraic understanding, was funded by the United States Institute of Education Sciences, and was conducted in US classrooms from 2019-2021.

The descriptions of the study and the resulting database are incredibly thorough. The authors have paid excellent attention to detail of the methods, measures, procedures, and resultant datasets from this study. The sample selection, inclusionary criteria, and the steps to arrive at the analytic sample are clearly described. Throughout the manuscript the authors provided direct links to detailed study components housed on their OSF repository (e.g., the algebra problems used in the FH2T). The authors have also provided an excellent depth of information on the measures. I appreciated the links to existing publications using these data, which will increase the probability that others will use this data for future studies. The section on ways for the study data to be reused is clear and comprehensive.

The repository is suitable for this subject, though is domain general. The data are not deposited under an open license due to constraints on the shared data with the school districts. The data are posted in a non-proprietary format. The data dictionary makes clear that the deposited data will be interpretable by third party users. The deposited data are actionable. The authors describe their adherence to ethical human subjects standards during the creation of this work, and the authors state that the data have been anonymized. Further the data are only accessible through a data sharing agreement.

Small Typo: "...the full data cannot be unrestricted due but is open to the..."